# Modified Atmosphere Packaging and 1-Methylcyclopropene Treatments Maintain the Fruit Quality of ‘Wonmi’ Persimmons during Export Simulation

**DOI:** 10.3390/foods11244004

**Published:** 2022-12-11

**Authors:** H. M. Prathibhani C. Kumarihami, Mi Hee Shin, Kyeong Eun Jang, Yun-Hee Kim, Kyeong Bok Ma, Jin Gook Kim

**Affiliations:** 1Department of Crop Science, Faculty of Agriculture, University of Peradeniya, Peradeniya 20400, Sri Lanka; 2Institute of Agriculture and Life Sciences, Gyeongsang National University, 501, Jinju-daero, Jinju-si 52828, Republic of Korea; 3Division of Applied Life Science, Graduate School of Gyeongsang National University, 501, Jinju-daero, Jinju-si 52828, Republic of Korea; 4Department of Biology Education, Gyeongsang National University, 501, Jinju-daero, Jinju-si 52828, Republic of Korea; 5Pear Research Institute, National Institute of Horticultural & Herbal Science, RDA, Naju-si 58216, Republic of Korea; 6Department of Horticulture, College of Agriculture and Life Science, Gyeongsang National University, 501, Jinju-daero, Jinju-si 52828, Republic of Korea

**Keywords:** fruit quality, MAP, 1-MCP, relative humidity, temperature

## Abstract

Persimmons are one of the most important export fruits in South Korea, where several tons are exported across the globe each year. In this study, the quality attributes of ‘Wonmi’ persimmon fruits were evaluated during an export simulation at 0 °C, 10 °C, and 24 °C with a combination of 1-Methylcyclopropene (1-MCP) and modified atmosphere packaging (MAP) treatments. The relative humidity during the export simulation was greater at room temperature (75–92%) and 0 °C (85% to 93%) than at 10 °C (42% to 60%). The results show that the application of 1-MCP and MAP treatments during the export simulation were effective in delaying the ripening of ‘Wonmi’ persimmons by reducing respiration and ethylene production. The suppressed expression of ethylene synthesis genes and cell wall modification genes reduced the ethylene production and maintain the fruit firmness, respectively. In addition, 1-MCP and MAP treatments were effective in maintaining SSC and color of ‘Wonmi’ persimmon fruits during the export simulation. Thus, by adopting these treatments, the overall quality of persimmon exports from South Korea can significantly improve.

## 1. Introduction

Persimmons (*Diospyros kaki* Thunb.) are one of the most economically important fruit crops in South Korea. The production of persimmons in Korea ranks second in the world next to China [1,2]. High-quality Korean sweet persimmons captivate global consumers owing to their high sugar content, crispy taste, small size, and rigorous quality control. South Korea has been exporting high-quality sweet persimmons to about fourteen countries, including Malaysia, Hong Kong, Singapore, Thailand, Dubai, the United States, and Canada, since its first export to Malaysia in 1993 [3,4]. About 8000 tons of persimmon fruits were exported in 2015, mostly to Southeast Asian countries including Malaysia, Hong Kong, Singapore, and Thailand [3,5]. The main producing area is in the southern part of Korea, especially in the Gyeongsangnam-do province, where 90% of the exported fruit originates [6]. The main persimmon variety exported is the non-astringent ‘Fuyu’ [3,5].

The continual supply of high-quality fruits to the export market is the focus of Korea’s long-term success in the competitive global export market. Korea has overcome its export competitiveness by growing newly improved persimmon cultivars, introducing farming and postharvest techniques for high-quality persimmons, and actively developing new markets. In 2014, the ‘Wonmi’ cultivar was introduced as an early-maturing, pollination-constant non-astringent (PCNA), high-quality, sweet persimmon cultivar derived from a cross between ‘Fuyu’ × ‘Taishu’ in Korea [6,7]. The average ‘Wonmi’ fruit weighs about 220 g, and its soluble solids content (SSC) average 15.1 °Brix. The fruit has a round oblate shape and an orange color skin with a graceful appearance. The juicy flesh has a pleasant taste and crispy texture. Physiological disorders, such as stylar-end or fine skin cracking, rarely occur in the ‘Wonmi’ cultivar [6,7].

However, being climacteric, persimmon fruits are characterized by autocatalytic ethylene biosynthesis, which makes them more perishable with a relatively short shelf life [1,2]. Rapid fruit softening during the postharvest period is a major challenge in the marketing and exporting of sweet persimmons. Recently, the ‘Wonmi’ persimmon was shipped to Taiwan by shipment in about ten days for the distribution schedule. The distribution showed softening fruits, demonstrating a loss in fruit quality.

Therefore, the fruit quality can diminish during the long-distance transport, and the shelf life is reduced during storage. A reduction in fruit-softening rates during shipping and storage would be beneficial, especially for distant markets. Therefore, there is an urgent need to develop simple and low-energy technology to protect ‘Wonmi’ sweet persimmons during export and to apply this technology in production practices. Therefore, the objective of the present study was to investigate the effects of modified atmosphere packaging (MAP) and 1-methylcyclopropene (1-MCP) treatments on maintaining postharvest quality and prolonging the storage time of ‘Wonmi’ sweet persimmons during an export simulation.

## 2. Materials and Methods

### 2.1. Plant Material and Simulation for Export

For our analysis, ‘Wonmi’ persimmon fruits were harvested on 8 October 2021 from 7-year-old trees grown in the persimmon orchard of the Pear Research Institute, RDA, Republic of Korea. Persimmons were picked at the commercial maturity stage. After harvest, fruits were transported to the fruit science and postharvest laboratory at Gyeongsang National University, Republic of Korea. The 166 fruits were graded according to color uniformity, absence of calyx damage, and surface defects. Persimmon fruits were divided into seven groups for the export simulation: (1) control (no treatment) distribution at room temperature (RT, 24 °C); (2) storage at 10 °C for 10 days + distribution at RT; (3) 1-MCP-treated fruit storage at 10 °C for 10 days + distribution at RT; (4) storage 0 °C for 10 days + distribution at RT; (5) 1-MCP-treated fruit storage at 0 °C for 10 days + distribution at RT; (6) MAP I: OPP anti-fogging film, 30 μm, OTR (oxygen transmission rate 80,000 CC), distribution at RT; and (7) MAP II: OPP anti-fogging film, 30 μm, OTR (oxygen transmission rate 160,000 CC), distribution at RT.

### 2.2. 1-MCP Treatments and Modified Atmosphere Packaging

Persimmon fruits were treated with 1-MCP on the same day after harvest. 1-MCP (EthylBloc, or SmartfreshTM, Bio-Technologies for Horticulture, Walterboro, SC, USA) in tablet form was directly dissolved in 100 mL of hot water (98 °C) in an airtight environment using a plastic tent. A treatment dose of 980 ppb (commercially used on persimmons) was applied in a hermetically sealed chamber with 4.2 g/3.8 m^3^. The duration of the treatment was 24 h at room temperature, and the control fruits were kept under the same conditions without the 1-MCP treatment.

For MAP storage, persimmon fruits were packaged into OPP anti-fogging film, 30 μm, 80,000 CC OTR (oxygen transmission rate), and 150,000 CC OTR. The OPP anti-fog film adds an anti-fog function for the film-like condensation deterrent produced by stretching polypropylene (PP) through the molten extrusion and cooling process as the main material and then stretching it to the one-axis or two-axis.

### 2.3. Weight Loss Rate

Fruit weight loss was measured on 10 fruits per treatment using a digital balance with an accuracy of 0.01 g (YP6001, YUKE Instrument, Shanghai, China). Weight loss was determined from the beginning to the end of the storage period and expressed as the percentage of weight loss relative to the initial weight.

### 2.4. Fruit Firmness

Firmness was measured using a rheometer (RHEO TEX SD-700, Sun Scientific Inc., Tokyo, Japan) fitted with an 8 mm round, flat probe that compressed the fruit to a depth of 3 mm with a crosshead speed of 120 mm·min^−1^. For flesh firmness, the fruit was sliced into longitudinal halves, and each half was measured in the central zone after the peel (~2 mm thick) was removed. The maximum force generated during penetration was recorded as firmness in Newton (N).

### 2.5. Soluble Solids Content (SSC)

The soluble solids content (SSC) for 10 biological replicates was measured using a hand refractometer (Pocket Refractometer, PAL-1, Atago Co., Ltd., Tokyo, Japan) calibrated in °Brix and expressed as a percentage (%).

### 2.6. Ethylene Production and the Respiration Rate

Ethylene production and respiration rate were determined by a gas chromatograph with three biological replicates. The individual fruit was weighed and placed in separate 2.1 L volume airtight polypropylene containers (HPL851-2.1L, Locknlock, Suzhou, China) fitted with a rubber septum for 2 h at room temperature (24 °C). Air samples of the headspace were removed from the septum with a syringe and injected into a gas chromatograph (GC-7890B; Agilent Technologies, Santa Clara, CA, USA) equipped with a stainless-steel column (2.0 m × 3.0 mm i.d.) packed with Porapak Q (Shinwa, Kyoto, Japan) and a flame ionization detector (FID) to measure the ethylene production. The respiration rate was evaluated using a gas chromatograph (GC 6890, Agilent Technologies, Santa Clara, CA, USA) equipped with a stainless-steel column (2.0 m × 3.0 mm i.d.) packed with Shincarbon ST (Shinwa, Kyoto, Japan) and a thermal conductivity detector (TCD). The results were expressed in mg·CO_2_ kg^−1^·hr^−1^ and μL·C_2_H_4_ kg^−1^·hr^−1^.

### 2.7. Color

The surface color measurements of the persimmon fruits were determined by two readings on two different symmetrical faces of the fruit in each replicate, using a colorimeter (CR-400, Minolta, Osaka, Japan). The colorimeter was calibrated to a standard white tile. The results were expressed in terms of Hue values and L*, a*, and b*, representing lightness, chromaticity on a green (−) to red (+) axis, and chromaticity on a blue (−) to yellow (+) axis, respectively. The 10 independent replicates were evaluated in each sampling time of each treatment.

### 2.8. RNA Isolation and Analysis of Gene Expression

The total RNA was isolated from the persimmon fruit using TRIzol reagent (Invitrogen, Carlsbad, CA, USA) and treated extensively with RNase-free DNase I to remove any contaminating genomic DNA. Quantitative real-time PCR analysis was performed using a Bio-Rad CFX96 thermal cycler (Bio-Rad) with EvaGreen fluorescent dye according to the manufacturer’s instructions. Linear data were normalized to the mean threshold cycle (Ct) of the actin reference gene. Gene-specific PCR primers are listed in Table 1.

### 2.9. Statistical Analysis

The experiment was conducted in a completely randomized design. The results were statistically analyzed with SPSS 17 (SPSS Inc., Chicago, IL, USA). The mean values of gene analysis data were compared by Tukey’s test at a 95% confidence level (*p* ≤ 0.05). Figures were made by Origin Pro 2021 (© OriginLab Corporation., Northampton, MA, USA).

## 3. Results

### 3.1. Change in Temperature and Relative Humidity during the Simulated Export Distribution Period

During the export simulation, the temperature and relative humidity were varied to reflect the changes inside the persimmon packaging boxes. The variations in temperature (Figure 1A) and relative humidity (Figure 1B) during the simulated export distribution period (10 days) at room temperature (24 °C), 10 °C, and 0 °C were recorded using data loggers. The temperature variations were slightly less at 10 °C compared to 24 °C (Figure 1A). During room-temperature storage at 24 °C, the temperature in the persimmon box dropped to nearly 20 °C due to a temperature reduction at end of October. The change in relative humidity during the simulated export distribution period showed great variation among the different treatments examined in this study (Figure 1B). The relative humidity of 92% was reduced to 75% in the persimmon packaging boxes stored at room temperature. The relative humidity at 10 °C was changed between 42% to 60%. At 0 °C, the relative humidity was maintained at 85% to 93%, while it was reduced to 70% over the storage time at room temperature. The modified atmospheric packaging stored at room temperature maintained a relative humidity of 97% to 99% for all 10 days of storage.

### 3.2. Change in the Weight Loss of the Sweet Persimmon ‘Wonmi’ during the Simulated Export Distribution Period

A gradual increase in weight loss was observed during the ripening of the ‘Wonmi’ fruit (Figure 2) during the simulated export distribution period. ‘Wonmi’ fruit without 1-MCP treatment stored at 10 °C for 10 days at 42–60% relative humidity showed 5% weight loss, while 1-MCP treated fruit stored in the same conditions exhibited 6% weight loss compared to fruit stored at room temperature, i.e., 24 °C (control; 4% weight loss). As a result of substantial weight loss, a severe loss in persimmon fruit quality was observed. At 10 °C, it was revealed that MAP packaging and relative humidity had a significant role in persimmon fruit quality maintenance compared to ordinary paper box packaging. ‘Wonmi’ fruit stored at room temperature, 24 °C, resulted in fruit weight loss of more than 5% after 10 days of storage. 1-MCP-treated ‘Wonmi’ fruit showed ≤5% weight loss after 6 days at 0 °C storage. Thus, 1-MCP treatment combined with cold storage at 0 °C is effective for maintaining fruit quality during the fruit-distribution channel. 

MAP recorded a reduction of 2% in weight loss during 14 days of storage at room temperature. The MAP containing 30 μm OPP anti-fogging film with 80,000 CC of oxygen transmission rate (OTR) was much more effective in reducing the weight loss rate of ‘Wonmi’ fruit compared to the 30 μm OPP anti-fogging film with 150,000 CC of OTR. After 10 days of storage at 10 °C, fruit softening, wrinkling, and faster water loss was observed in untreated persimmon fruits than in 1-MCP-treated fruits. In addition, fruits stored at 10 °C lost their commercial quality and were not measured after 10 days of storage. In addition, fruits without 1-MCP treatment stored at 0 °C were not tested after 3 days of storage at room temperature due to quality loss.

### 3.3. Change in the Firmness during the Simulated Export Distribution Period

‘Wonmi’ fruits stored at room temperature (24 °C) reduced their firmness after 14 days of storage up to 37 N from an initial firmness of 51 N at harvest (Figure 3). A faster firmness loss from 51 N to 29 N within 10 days of storage was recorded for the persimmons stored at 10 °C without 1-MCP treatment. In contrast, the 1-MCP-treated persimmons stored at 10 °C were significantly effective in reducing the rate of firmness loss within 10 days of storage (52 N up to 48 N). Although cold storage at 0 °C was effective in maintaining fruit firmness, a rapid loss of firmness up to 21 N was observed in 1-MCP-untreated ‘Wonmi’ fruit three days after they were moved to room-temperature storage. However, 1-MCP-treated persimmon fruits retained their firmness efficiently at 0 °C storage, and a reduction up to 40 N was recorded fourteen days after they were moved to room-temperature storage. Persimmon fruit stored at room temperature using MAP with 80,000 CC of OTR had a firmness of 45 N after 14 days of storage, while this was 37 N for the MAP with 150,000 CC of OTR. 1-MCP-treated fruits stored at 0 °C for 10 days followed by room-temperature storage were effective in maintaining firmness compared to the untreated fruits stored at similar storage conditions. The 1-MCP-treated fruits had a firmness of 40 N after 14 days of storage, while the untreated fruits exhibited only 21 N at 3 days after storage. Therefore, low-temperature storage alone is not sufficient for fruit firmness maintenance during the distribution channel; a combination of 1-MCP treatment and low-temperature storage is most effective in fruit firmness maintenance.

### 3.4. Change in the Soluble Solids Content (SSC) of the ‘Wonmi’ Persimmon during the Simulated Export Distribution Period

At harvest, the ‘Wonmi’ persimmons had 14.7% of SSC. Irrespective of the different storage temperatures during the simulated export distribution, the SSC increased with time (Figure 4). The ‘Wonmi’ fruit stored (both 1-MCP-treated and -untreated fruits) at 10 °C showed a rapid increase of SSC during 10 days of storage. The SSC of 1-MCP-treated ‘Wonmi’ fruits stored at 0 °C increased slowly. At the end of the simulation export period, the greatest SSC of 16.1 % was recorded for the 1-MCP-treated fruit stored at 0 °C for 10 days followed by room-temperature storage for 14 days. The SSC of persimmon fruit stored at room temperature using MAP with 80,000 CC of OTR was nearly constant during the storage period.

### 3.5. Changes in the Skin Color of the Sweet Persimmon ‘Wonmi’ during the Simulated Export Distribution Period

The skin color of ‘Wonmi’ persimmons changes from green to orange as the fruit matures. The L*, b*, and hue values of ‘Wonmi’ persimmon fruit skin gradually decreased, while the a* value gradually increased as storage time increased regardless of the storage conditions applied for the simulated export distribution (Figure 5). The highest L* value of 66 (a measure of the brightness of the color) was obtained for the fruit skin at harvest (Figure 5A) and was reduced with the storage time, meaning that the fruit developed darker skin color. The reduction rate of the L* value was lower for the 1-MCP-treated fruit stored at 0 °C for 10 days followed by room-temperature storage (Figure 5A). ‘Wonmi’ fruit stored at 24 °C (control fruit) resulted in lower L* values than those stored at cold temperatures using MAP, indicating that persimmons stored at low temperatures (0 °C and 10 °C) are brighter than those stored at high temperatures (24 °C). The redness of the ‘Wonmi’ persimmon fruit was measured as the a* value and had an initial value of 10 (Figure 5B). The a* values of those treated with 1-MCP and stored at 0 °C for 10 days followed by room-temperature storage had lower a* values than others. The b* values for persimmon fruit skin color were reduced slightly during storage regardless of the storage conditions (Figure 5C). A higher b* value was reported at the harvest and was reduced with storage, possibly because beta-carotenes (pale-yellow color) in the fruit skin reach their highest concentration before full ripening, whereas lycopene (red color) and β-carotene (orange color) achieve their peaks during fruit ripening. 1-MCP-treated fruit stored at 0 °C for 10 days followed by room-temperature storage had paler or brighter orange skin color (higher hue values), while fruits subjected to other storage conditions were a deep orange (lower hue values, Figure 5D). 

### 3.6. Changes in the Ethylene Production Rate of the ‘Wonmi’ Persimmon during the Simulated Export Distribution Period

‘Wonmi’ persimmons stored at room temperature showed a rapid increase in postharvest ethylene production following a pattern of decreasing and increasing during storage (Figure 6). The production of ethylene was delayed by 1-MCP treatment for 6 days at room temperature after 0 °C storage and resulted in an increase in ethylene production in the later stages of storage. 

After 14 days of storage at room temperature, a postharvest ethylene production of 4.9 μL·kg^−1^·hr^−1^ was reported for the ‘Wonmi’ fruit stored using MAP with 80,000 CC of OTR, which was similar to the ethylene production of 1-MCP-treated fruits (4.6 μL·kg^−1^·hr^−1^). Fruit stored using MAP with 150,000 CC of OTR showed a slightly greater ethylene production of 5.9 μL·kg^−1^·hr^−1^ after 14 days of storage at room temperature compared to other treatments. 

### 3.7. Changes in the Respiration Rate of ‘Wonmi’ Persimmons during the Simulated Export Distribution Period

The respiration rate of ‘Wonmi’ persimmons stored at room temperature gradually increased until seven days after harvest (Figure 7). Thereafter, those fruits showed a rapid decrease in respiration rate followed by a rapid increase at 11 days after harvest. 1-MCP-treated persimmons showed a lower rate of respiration during storage at 10 °C for 10 days compared to untreated fruits in the same storage conditions. The fruits without 1-MCP treatment stored at 0 °C for 10 days followed by room-temperature storage also showed a rapid increase in respiration rate after moving to room-temperature storage. In contrast, 1-MCP-treated fruits stored at 0 °C for 10 days followed by room-temperature storage recorded gradual changes in respiration rate during their storage period.

The concentrations of oxygen (O_2_) and carbon dioxide (CO_2_) were measured in the MAP after 14 days of harvest. MAP with 80,000 CC of OTR recorded a 15.6% O_2_ concentration and 0.6% CO_2_ concentration. The O_2_ and CO_2_ concentrations in MAP with 150,000 CC of OTR were 17.6% and 0.3%, respectively. OTR 80,000 CC MAP packaging treatment has been determined to have contributed to the suppression of negligence and quality maintenance by maintaining high humidity (97–99%) and hypoxic, high CO_2_ conditions for 14 days of storage compared to other storage treatments. The accumulation of O_2_ and CO_2_ gases in ‘Wonmi’ persimmons stored at room temperature and the fruits stored at 0 °C for 10 days followed by room-temperature storage for 3 days were compared based on their respiration rates.

### 3.8. Expression of Ethylene Biosynthesis-, and Cell Wall Modification-Related Genes of the Sweet Persimmon ‘Wonmi’ during the Simulated Export Distribution Period

The expressions of four ethylene biosynthesis-related genes (*ACS*, *ACO*, *ETR*, and *ERS*) and six cell wall modification-related genes (*β-Gal*, *PG*, *Egase*, *PL*, *PE*, and *XTH*) were studied during postharvest storage of ‘Wonmi’ persimmons (Figure 8 and Figure 9).

A significant difference in the *ACS* and *ACO* genes was not observed between harvest day, 7 days, and 10 days of storage at 24 °C. After 14 days of storage at 24 °C, the *ACS* and *ACO* levels were 10-fold and 2-fold higher compared with the harvest day, respectively (Figure 10). The relative expression of the *ETR* gene was higher at 10 days of storage followed by 14 days of storage at 24 °C (Figure 10). *ERS* gene expression was higher at 7, 10, and 14 days of storage.

The cell wall modification-related genes, including *β-Gal*, *PG*, *Egase*, *PE*, and *XTH*, were highly expressed after 14 days of storage at 24 °C, while the *PL* gene was highly expressed at 7 and 10 days of storage.

‘Wonmi’ persimmons stored at 0 °C for 10 days followed by room-temperature storage at 24 °C for 3 days showed the highest expressions in ethylene biosynthesis-related genes (*ACS*, *ACO*, *ETR*, and *ERS*) (Figure 10) and cell wall modification-related genes (*β-Gal*, *PG*, *Egase*, *PL*, *PE*, and *XTH*) compared to the 1-MCP-treated fruits stored in the same conditions (Figure 11). 

## 4. Discussion

### 4.1. Effects of Temperature and Relative Humidity on ‘Wonmi’ Persimmon Fruit Quality Attributes during the Simulated Export Distribution Period

Storage temperature and relative humidity are crucial in determining the shelf life of persimmons [8,9]. Controlling the ripening and shelf life of persimmon fruit is primarily based on controlling ethylene production and activity. The optimum storage temperature and relative humidity retard the metabolic processes while reducing the respiration and ethylene production rates and reducing water loss. Selecting the correct cold storage conditions and duration is essential to avoid the loss of fruit quality during prolonged transportation and fruit shipment to overseas markets [9]. All fresh produce can be kept to its maximum shelf life under its optimum storage conditions, including temperature and relative humidity.

The present results show that different temperature regimes (24 °C, 10 °C, and 0 °C) changed the relative humidity during the simulation period of persimmon fruit distribution. The relative humidity was greater at room temperature (75–92%) and 0 °C (85% to 93%) than at 10 °C (42% to 60%). Iqbal et al. [9] also observed changes in relative humidity at various persimmon storage temperatures, namely 10 °C, 20 °C, and 30 °C, with a relative humidity of 89%, 79%, and 69%, respectively. Persimmon fruit quality was greatly influenced by the storage temperature, and the fruit stored at 10 °C with 89% relative humidity maintained overall quality attributes (firmness, SSC, titratable acidity—TA, weight loss, color, taste, pH, ascorbic acid content, and moisture content) up to 36-day storage durations [9]. However, the ‘Wonmi’ persimmon stored at 10 °C showed fruit softening, wrinkling, and faster water loss. These persimmons lost their commercial quality and were not measured after 10 days of storage. This can be attributed to the 42% to 60% lower relative humidity in the 10 °C simulated export distribution. Low relative humidity can cause a greater vapor deficit, which accelerates fruit weight loss [10]. Tu et al. [10] found that higher relative humidity (65 and 95%) maintained firmness and reduced weight loss in ‘Braeburn’ and ‘Jonagold’ apples under retail shelf temperature (20 °C). ‘Rojo Brillante’ persimmons stored at 1 °C (85–90% RH) presented lower weight losses than fruit stored at 15 °C (85–90% RH) [11].

Moreover, the optimal fruit storage and transport temperature is a critical factor that determines the quality of the final fruit. The fruit quality of persimmons was greatly influenced by the physiological stage of the fruit and the storage temperature during long-distance shipment [8,11]. Although refrigerated transport is important to be considered for transporting persimmon fruit, the storage period of persimmon fruit is limited under low temperatures and reduces its commercial value. Many tropical/sub-tropical horticultural crops, e.g., avocados (*Persea americana* Mill.), bananas (*Musa* spp.), mangos (*Mangifera indica* L.), and tomatoes (*Lycopersicon esculentum* Mill.), are susceptible to chilling injury after exposure to very low but not freezing temperatures. Chilling injury can be a major cause of deterioration of persimmons during marketing. The main symptom of chilling injury is a rapid loss of flesh firmness, resulting in severe fruit softening, which undoubtedly affects final fruit quality [8,11,12,13,14]. Besada et al. [8] revealed that ‘Triumph’ persimmons, one of the main persimmon cultivars in Spain, must be shipped in refrigerated containers at 1 °C and posterior transport in refrigerated trailers at 1 or 3 °C to ensure fruit quality in the retail market. They reported that ‘Triumph’ persimmons transported at 9 °C caused an important fruit quality loss due to significant flesh softening. The cultivar, storage temperature, and duration will affect the incidence and severity of the chilling injury [13]. ‘Fuyu’ and other non-astringent cultivars are chilling-sensitive at temperatures between 5 °C and 15 °C and will exhibit flesh browning and softening [8,12,13]. Transporting ‘Jiro’ persimmons at 2 °C prevents loss of fruit quality during distribution compared to 15 °C [12]. Therefore, it is very important to choose an appropriate transport and storage temperature considering the chilling sensitivity of persimmon fruits [12]. The recommended storage and transport temperature for persimmons is 0 °C at a relative humidity of 90–95% [8,13].

### 4.2. Effects of 1-MCP Treatment on ‘Wonmi’ Persimmon Fruit Quality Attributes during the Simulated Export Distribution Period

Persimmons are very sensitive to ethylene, and storing them at 0 °C alone is not efficient in maintaining fruit quality. Therefore, the transportation of fruits to export markets requires appropriate technology combined with low-temperature storage to maintain fruit quality and freshness. ‘Wonmi’ persimmons stored at 0 °C showed a rapid loss of firmness up to 21 N three days after they were moved to room-temperature storage, which may be attributed to the ethylene sensitivity of persimmon fruits. Thus, the removal and/or exclusion of ethylene gas from transport and storage facilities is essential. Currently, the application of 1-MCP before cold storage at 0 °C is highly recommended to maintain fruit quality and extend the shelf life of persimmons [8,13]. 1-MCP is a potent ethylene antagonist: an inhibitor of ethylene action. 1-MCP irreversibly binds to ethylene receptors in plant tissues, blocking the binding of ethylene, thereby blocking ethylene action in the tissues [1,4,15,16]. In the present study, we observed that the incorporation of 1-MCP during postharvest prolongs the low-temperature storage capacity of persimmons. The fruit weight loss increased mainly due to respiration and transpiration along with several metabolic processes. The weight loss of the ‘Wonmi’ persimmons increased as the storage period advanced. However, 1-MCP treatment significantly lowered the weight loss of persimmon fruits during storage. The low weight loss in 1-MCP-treated fruits can be attributed to a low level of respiration rate and maintaining the firmness of the fruit while reducing the water losses through transpiration.

One of the main quality attributes used to assess the fruit-ripening progress is fruit firmness, which directly affects consumer acceptance. During ripening, the cell walls and intercellular adhesives are degraded, and the fruit tissue becomes soft. In the present study, the firmness of ‘Wonmi’ persimmons gradually decreased as storage time increased. Furthermore, since 1-MCP is an ethylene antagonist, the effect of 1-MCP on fruit firmness may be caused by a decrease in the rate of ethylene production. Ethylene is significantly involved in fruit softening and affects overall fruit ripening as persimmons are a climacteric fruit. Moreover, delayed fruit softening in 1-MCP-treated ‘Wonmi’ persimmons was associated with reduced expression of ethylene biosynthesis-related genes (*ACS*, *ACO*, *ETR*, and *ERS*) and cell wall modification-related genes (*β-Gal*, *PG*, *Egase*, *PL*, *PE*, and *XTH*). Storing 1-MCP-treated ‘Wonmi’ persimmons at 10 °C was significantly effective in reducing the rate of firmness loss within 10 days of storage (52 N up to 48 N), while untreated fruits rapidly lose their firmness from 51 N to 29 N within 10 days of storage at 10 °C. Furthermore, 1-MCP-treated persimmons retained their firmness efficiently at 0 °C storage, and a reduction up to 40 N was recorded fourteen days after they were moved to room-temperature storage. Karmoker et al. [16] concluded that 1-MCP can suppress the ethylene sensitivity and maintain the firmness of Japanese persimmon fruits at 25 °C (slightly) and 0 °C (strongly) for 1 and 8 weeks, respectively. 1-MCP has proven to be a potent inhibitor of ethylene action and is effective in reducing fruit firmness loss in other persimmon cultivars such as ‘Fuyu’ [4], ‘Rojo Brillante’ [14], and ‘Bansi’ [17]. 1-MCP affects ethylene biosynthesis by regulating the expression of 1-aminocyclopropane-1-carboxylic acid synthase (*ACS*) and 1-aminocyclopropane-1-carboxylic acid oxidase (*ACO*) genes in the ethylene synthesis pathway. He et al. [18] reported that twelve cell-wall-modifying genes (*DkPG1*, *DkPL1*, *DkPE1/2*, *Dkβ-GAL1*, *DkEGase1*, *DkXTH2/9/10/11*, *DkMAN1*, and *DkEXP4*) and four ethylene biosynthesis genes (*DkACS1/2* and *DkACO1/2*) were induced by ethylene and suppressed by 1-MCP during persimmon fruit storage.

The present study also revealed the effectiveness of 1-MCP treatment combined with the low-temperature storage at 10 °C and 0 °C in reducing respiration and ethylene production rates, thereby maintaining fruit weight, firmness, SSC, and color. A delay in ethylene production by 1-MCP treatment may be due to 1-MCP irreversibly blocking ethylene binding sites and inhibiting ethylene autocatalytic production. Similar reports of delayed ethylene production with the application of 1-MCP have been documented in other persimmon cultivars [17,18]. SSC is important in the sensory quality of sweet persimmons. In line with Karmoker et al. [16], the SSC of 1-MCP-treated ‘Wonmi’ fruit stored at 0 °C was maintained during the simulated export period, indicating that 1-MCP treatment can delay the ripening of stored persimmon fruit.

1-MCP was effective in delaying fruit softening; changing SSC, TA, and color; and significantly extending the shelf life of ‘Nathanzy’ persimmons stored at 20 °C during 30 days of storage [15]. Suppression of fruit respiration and ethylene production by 1-MCP was reported to maintain fruit firmness, color, and SSC in ‘Mopan’ persimmons stored at 22 °C compared to untreated fruit [1]. Preventing skin color changes in fruit during storage is also very important. Similar to our results, 1-MCP has been reported to maintain the color of Japanese persimmons at both 25 °C and 0 °C [16].

### 4.3. Effects of MAP Treatment on ‘Wonmi’ Persimmon Fruit Quality Attributes during the Simulated Export Distribution Period

Moreover, MAP combined with low-temperature storage has been widely used to maintain fruit quality and extend its shelf life during long-term storage and marketing. MAP is an efficient method to avoid chilling injury [12]. The desired storage atmosphere of MAP is obtained by reducing O_2_ concentration and increasing CO_2_ concentration in a sealed polymer film package compared with that in ambient air [12,19]. MAP with 30 μm OPP anti-fogging film with 80,000 CC of OTR used in the simulation export distribution of ‘Wonmi’ persimmons maintained a relative humidity of 97% to 99% and maintained hypoxic, high-carbon-dioxide conditions, thereby maintaining the quality attributes of ‘Wonmi’ persimmons during the export distribution simulation. MAP with a 40 μm thick LDPE membrane was effective in maintaining low O_2_ and high CO_2_ inside and inhibited the respiration rate, which resulted in a decrease in the number of softened fruits in ‘Jiro’ persimmons [12]. In addition to the low O_2_ created inside the package, MAP also created high humidity [12]. MAP is a promising way to slow down the respiration rate, ethylene production, and water loss, thus reducing the metabolic processes that lead to the ripening and quality deterioration of fresh produce. ‘Wonmi’ fruit stored in MAP with 80,000 CC of OTR showed an ethylene production like that of 1-MCP-treated fruits (4.6 μL·kg^−1^·hr^−1^). The application of MAP affected the activity of ethylene synthesis, as the low O_2_ in the package inhibited ACC oxidase activity, leading to reduced ethylene production [12]. The effectiveness of MAP in the reduction of weight loss and flesh softening of ‘Wonmi’ persimmons was observed. Additionally, MAP was effective in maintaining the SSC and fruit color in ‘Wonmi’ fruit, thereby delaying fruit ripening during the export delivery simulation. Fahmy and Nakano [12] reported that ‘Jiro’ persimmons stored in MAP did not produce detectable levels of ethylene and delayed fruit ripening and softening due to high relative humidity conditions. Ahmed et al. [20] demonstrated that ‘Hachia’ and ‘Triumph’ persimmon fruits packaged in MAP using 7 μm low-density polyethylene (LDPE) bags retain satisfactory quality for 90 days at 0 °C (85–90% RH) following 7 days at 20 °C (85–90% RH) for ripening. Most ‘Fuyu’ persimmon fruits are stored and marketed using MAP basically to delay the softening of the fruit [20]. According to our results, storage at low temperatures such as 0 °C and 10 °C combined with 1-MCP and MAP treatments was significantly effective in maintaining the skin color of persimmons during the simulated export distribution. The differences in these values are likely due to differences in storage temperature, type of packaging method (MAP), and application of 1-MCP.

## 5. Conclusions

In this study, we examined the various factors affecting ‘Wonmi’ persimmon fruit quality degradation during simulated export distribution to find the optimal conditions for storage. We analyzed various persimmon fruit quality attributes such as weight loss, firmness, SSC, and fruit skin color. Our results reveal that ‘Wonmi’ persimmons treated with 1-MCP, stored at 0 °C, and packaged with MAP maintained the best quality during simulated export distribution. These results match well with existing studies on persimmon fruit distribution and can significantly improve the overall quality of ‘Wonmi’ persimmon exports in the future.

## Figures and Tables

**Figure 1 foods-11-04004-f001:**
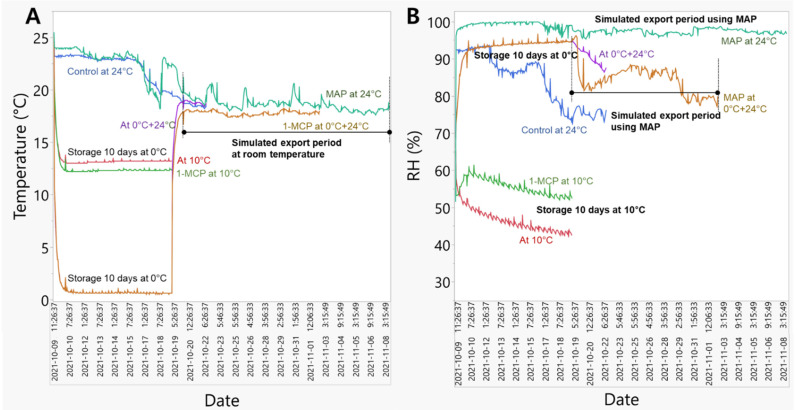
The change of temperature (**A**) and relative humidity (**B**) during the simulated export distribution of ‘Wonmi’ persimmons (1-MCP; 1-Methylcyclopropene, MAP; modified atmosphere packaging).

**Figure 2 foods-11-04004-f002:**
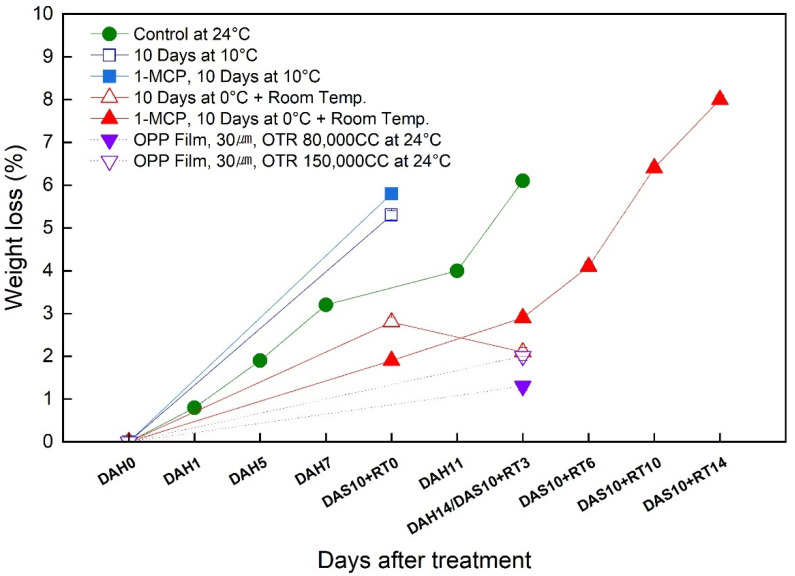
Change in the weight loss rate of the ‘Wonmi’ persimmon during the simulated export period. DAH, days after harvest; DAS, days after storage; RT, room temperature; OTR, oxygen transmission rate.

**Figure 3 foods-11-04004-f003:**
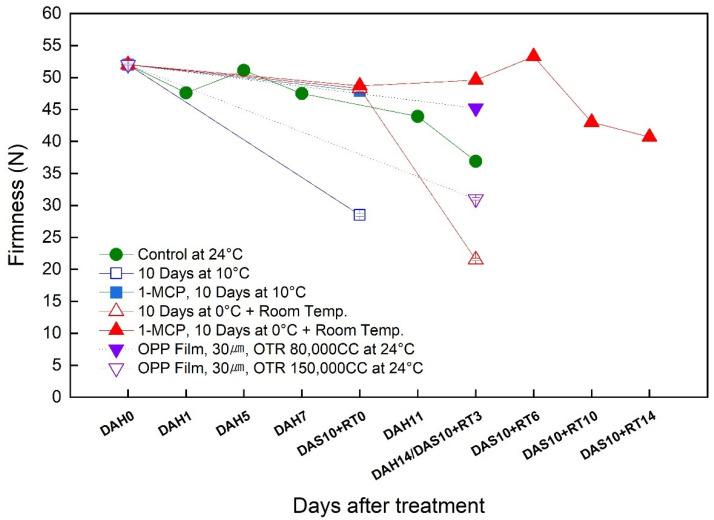
Change in firmness of ‘Wonmi’ persimmons during the simulated export period. DAH, days after harvest; DAS, days after storage; RT, room temperature; OTR, oxygen transmission rate.

**Figure 4 foods-11-04004-f004:**
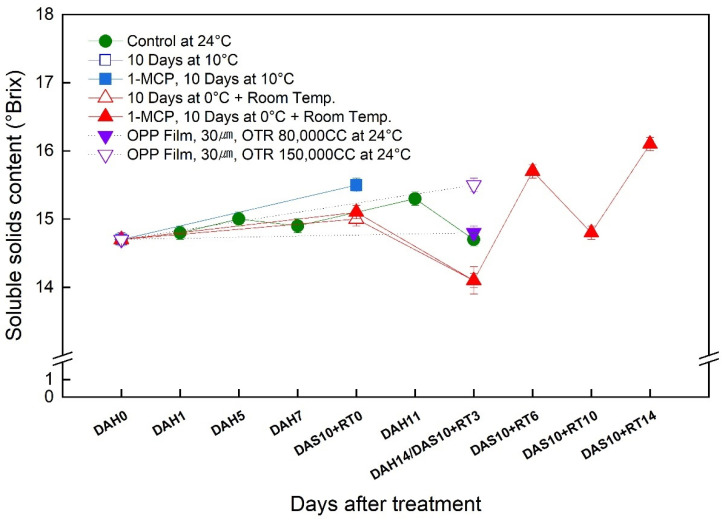
Change in the content of soluble solids of ‘Wonmi’ persimmons during the simulated export period. DAH, days after harvest; DAS, days after storage; RT, room temperature; OTR, oxygen transmission rate.

**Figure 5 foods-11-04004-f005:**
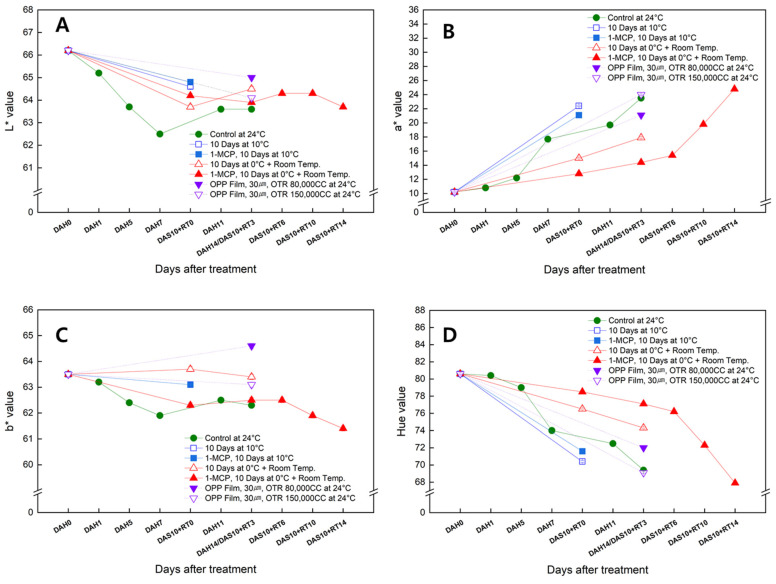
Changes in fruit color; L* value (**A**), a* value (**B**), b* value (**C**), and hue (**D**) in the ‘Wonmi’ persimmon during the simulated export period. DAH, days after harvest; DAS, days after storage; RT, room temperature; OTR, oxygen transmission rate.

**Figure 6 foods-11-04004-f006:**
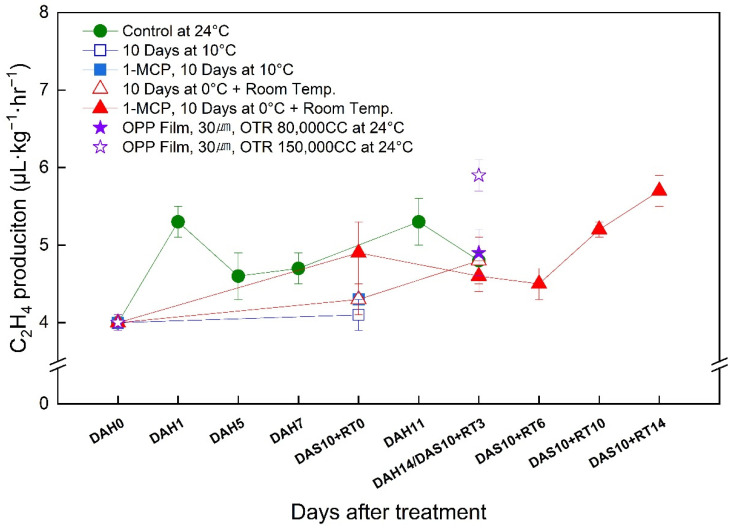
Change in the ethylene production rate of ‘Wonmi’ persimmons during the simulated export period. DAH, days after harvest; DAS, days after storage; RT, room temperature; OTR, oxygen transmission rate.

**Figure 7 foods-11-04004-f007:**
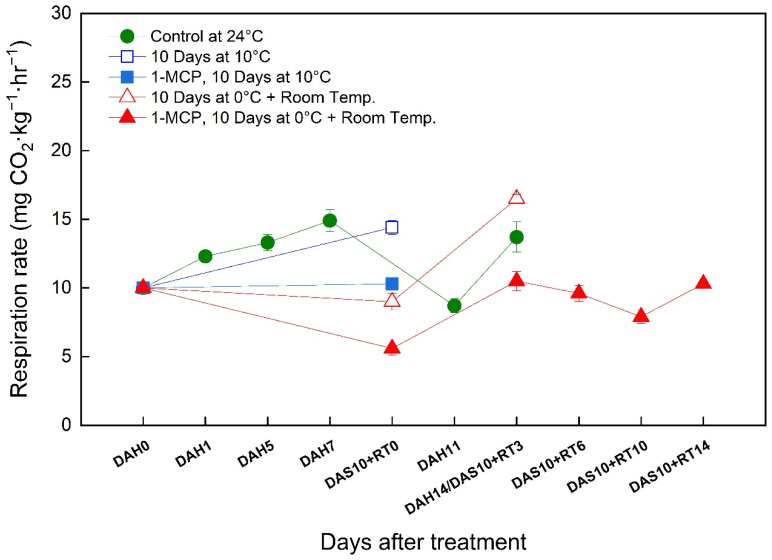
Change in respiration rate of ‘Wonmi’ persimmons during the simulated export period. DAH, days after harvest; DAS, days after storage; RT, room temperature; OTR, oxygen transmission rate.

**Figure 8 foods-11-04004-f008:**
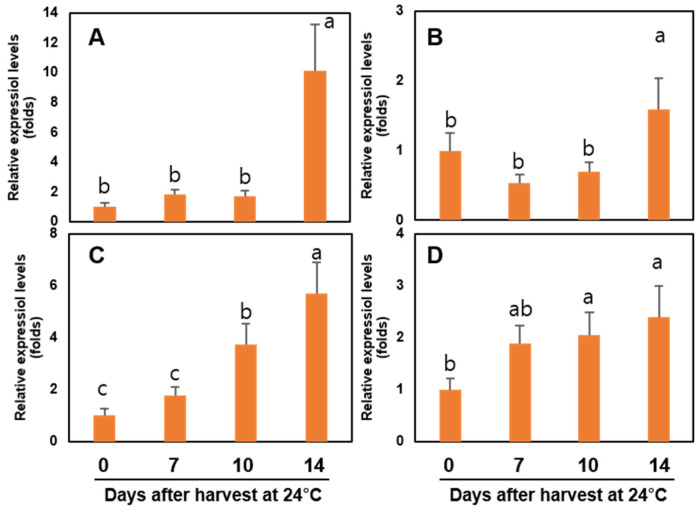
Ethylene-related genes expressions: (**A**) *ACS* (ACC synthase), (**B**) *ACO* (ACC oxidase), (**C**) *ETR* (ethylene receptor), and (**D**) *ERS* (ethylene response sensor) in ‘Wonmi’ persimmons during postharvest storage at 24 °C. Values followed by different lowercase letters are significantly different; Tukey’s test at *p* ≤ 0.05.

**Figure 9 foods-11-04004-f009:**
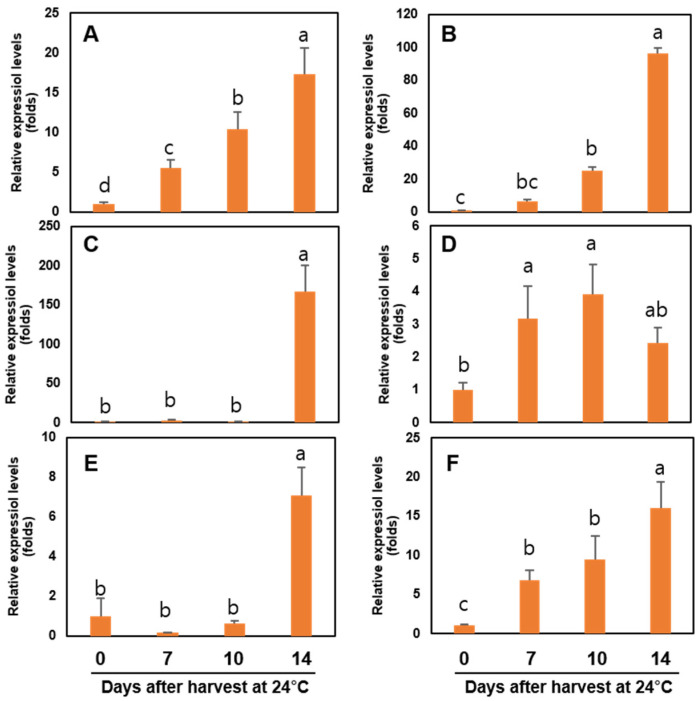
Cell wall modification-related genes: (**A**) *β-Gal* (beta-galactosidase), (**B**) *PG* (polygalacturonase), (**C**) *Egase* (endo-1,4-beta-glucanase), (**D**) *PL* (pectate lyase), (**E**) *PE* (pectinesterase), and (**F**) *XTH* (xyloglucan endo-transglucosylase/hydrolase) in ‘Wonmi’ persimmons during postharvest storage at 24 °C. Values followed by different lowercase letters are significantly different; Tukey’s test at *p* ≤ 0.05.

**Figure 10 foods-11-04004-f010:**
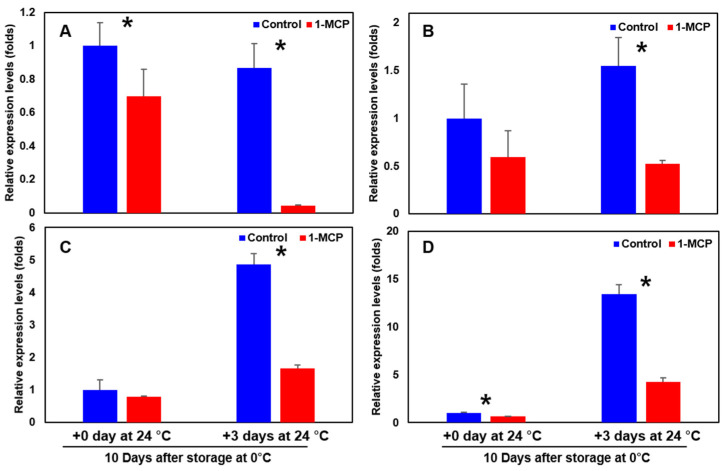
Ethylene-related genes: (**A**) *ACS* (ACC synthase), (**B**) *ACO* (ACC oxidase), (**C**) *ETR* (ethylene receptor), and (**D**) *ERS* (ethylene response sensor) in ‘Wonmi’ persimmons after 1-MCP treatment 10 days after postharvest storage at 0 °C; *, indicates significant differences between treatments, according to the *t*-test at *p* ≤ 0.05.

**Figure 11 foods-11-04004-f011:**
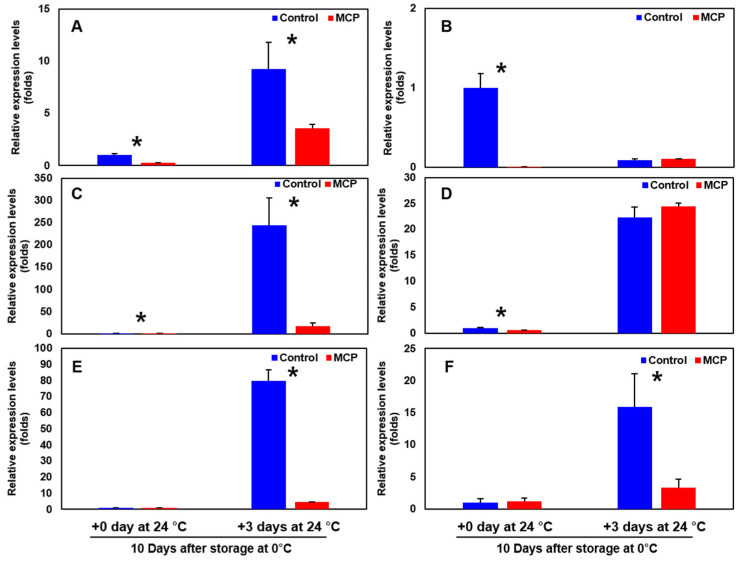
Cell wall modification-related genes: (**A**) *β-Gal* (beta galactosidase), (**B**) *PG* (polygalacturonase), (**C**) *Egase* (endo-1,4-beta glucanase), (**D**) *PL* (pectate lyase), (**E**) *PE* (pectinesterase), and (**F**) *XTH* (xyloglucan endo-transglucosylase/hydrolase) of 1-MCP-treated ‘Wonmi’ persimmons after10 days after postharvest storage at 0 °C; *, indicates significant differences between treatments, according to the *t*-test at *p* ≤ 0.05.

**Table 1 foods-11-04004-t001:** The primer sequences and gene information used in this study.

Gene Name	Forward Primer	Reverse Primer
*ACS*	*AGAATCCGGACGTTCGTGGATGA*	*AAGCATAGGGGAGTGAGGCGACAAC*
*ACO*	*TGGCAATGATGCTGTTATCTATC*	*CGAACTATTACAAATAACATGTGTC*
*ERS*	*GCGGATCTCATTGGACATAAGC*	*GAGCTTAAGCTTACAGGCACAC*
*ETR*	*AGATGCCTGCAGATTGGCATGA*	*GCAAGGCTGTGTTCACCATGGC*
*β-Gal*	*TACCAAACTTACCGGCTTGG*	*TTAGCCCCCTCTACCTTCGT*
*EGase*	*TCATGATATTGAACACCTCTGG*	*CAACACTTCAACAGCCCTACA*
*PE*	*ATTACATCGATTGTGGGTTTTG*	*TGCATGCATATTTACCATCAA*
*PG*	*GCCCCATTGTCCTTGAATCAGAGA*	*AAATGCACCACACTACGCTT*
*PL*	*CTTTGAATTTTCCCCGGCCC*	*TGCTATCTCGGGGCCAAAAG*
*XTH*	*ACGCCAAGTTCTGCGACAC*	*GGGTATCGCTTCCTGTCG*
*Actin*	*TTGGAGCAGAGAGATTCCGC*	*TTGCTCATCCGGTCAGCAAT*

## Data Availability

All data are included in the manuscript.

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
