# Peer review of "Modified Atmosphere Packaging and 1-Methylcyclopropene Treatments Maintain the Fruit Quality of ‘Wonmi’ Persimmons during Export Simulation"

_foods, 2022, doi:10.3390/foods11244004_

Round 1

Reviewer 1 Report

The paper presents a modified atmosphere packaging and 1-methylcyclopropene 2 treatments to maintain the export persimmon quality. Comparing 1-MCP and MAP in different temperatures, humidity and evaluating fruit firmness, soluble solids content, ethylene production, respiration rate, color and RNA isolation. The experimental results show that this special treatment method can effectively maintain the product quality, hardness and etc. under the same temperature and humidity. Thus, this work is expected to achieve some success in food packaging. However, there are still some points worth discussing. My detailed comments are as follows:

1.    One scheme to show the experimental procedure is suggested to better understand this work for readers.

2.    In Fig. 1, the horizontal and vertical coordinates are not clearly indicated. Please indicate the specific meanings of a, b, etc. in Fig. 8 and Fig. 9.

3.    All the figures should be modified to have a better readability and resolutions, especially the texts.

4.    Typically, coatings gave the anti-browning effect and improved the appearance of fruits. Please supplement the comparison pictures of natural conditions and processed samples at different time intervals.

5.    Please supplement antibacterial performance of this treatment against Staphylococcus aureus or Escherichia coli and other common bacteria.

6.    Please design experiment to describe the oxygen permeability and water vapor barrier of OPP anti - fogging film.

7.    There are still some formatting errors and syntax errors that need to be carefully corrected. In addition, more recent references should be added.

8.    More background on the packaging materials should be provided with supporting articles: Packaging and degradability properties of polyvinyl alcohol/gelatin nanocomposite films filled water hyacinth cellulose nanocrystals; Nanomaterials 12 (18), 3158, 2022; Development and characterization of food packaging bioplastic film from cocoa pod husk cellulose incorporated with sugarcane bagasse fibre; etc.

Author Response

Dear Reviewer 1

Thank you for your valuable comments.

Please, confer the revised manuscript.

Reviewer 2 Report

1. What is the reason for choosing 10 days? Some persimmons can be stored for up to 50 days, and some storage tests can even be up to 70 days.

2. Lack of references to experimental methods

3. The packaging of the final sample should have a legend

4.Although this study has the effect of revealing the linked use of 1-MCP and MAP, many references are cited in the text. However, the research on the application of A-MCP and MAP to persimmons is very diverse. The results of this research, I personally think that the innovation is not enough, and there is a lack of a more in-depth mechanism discussion (after all, the conditions of 1-MCP and MAP are fixed)

In conclusion, authors must clearly present a "research innovation" or "technological breakthrough" based on existing research

Author Response

Dear Reviewer 2

Thank you for your valuable comments.

Please, confer the revised manuscript.

Round 2

Reviewer 1 Report

Authors have addressed most of the issues well except one problem. In introduction section, author provided much introduction on the fruit, however, the background on the packaging issue to this fruit is missing. Authors should also give introduction on the present packaging strategies for this fruit, and provide their advantages and disadvantages. The following recent and relevent articles could be considered during the introduction: Packaging and degradability properties of polyvinyl alcohol/gelatin nanocomposite films filled water hyacinth cellulose nanocrystals; Nanomaterials 12 (18), 3158, 2022; Development and characterization of food packaging bioplastic film from cocoa pod husk cellulose incorporated with sugarcane bagasse fibre; etc.

Author Response

Thank you so much for your valuable comment. Please, confer the cover letter.

Reviewer 2 Report

The author seems to have provided some explanations, but it still needs to be clarified. For example, the 10-day simulated export procedure or the simulated export target location are also suggested to have relevant instructions. After all, the time spent exporting from South Korea to Europe should be different from exporting to Japan. .

Author Response

(The authors gave the same response as above.)
